# The Effect of Imidacloprid on the Volatile Organic Compound Profile of Strawberries: New Insights from Flavoromics

**DOI:** 10.3390/foods12152914

**Published:** 2023-07-31

**Authors:** Ning Yue, Hongping Wang, Chunmei Li, Chen Zhang, Simeng Li, Jing Wang, Fen Jin

**Affiliations:** 1Key Laboratory of Agro-Product Quality and Safety, Institute of Quality Standards & Testing Technology for Agro-Products, Chinese Academy of Agricultural Sciences, Beijing 100081, China; 2Institute of Food Science and Technology, Chinese Academy of Agricultural Sciences, Beijing 100193, China

**Keywords:** volatile organic compounds, strawberry, organic agriculture, flavoromics, imidacloprid

## Abstract

Organic agriculture is of great socioeconomic significance because it can promote the nutritional quality of horticultural crops and is environmentally friendly. However, owing to the lack of techniques for studying complex aroma-related chemical profiles, limited information is available on the influence of organic practices on the flavor quality of strawberries, one of the primary factors driving consumer preferences. Here, two-dimensional gas chromatography combined with time-of-flight mass spectrometry (GC×GC-TOF-MS) and flavoromics analysis was employed to investigate the profiles and differences in the volatile organic compounds (VOCs) of strawberries under organic (without imidacloprid) and conventional (with imidacloprid) agricultural practices. A total of 1164 VOCs, representing 23 chemical classes (e.g., aldehydes, terpenes, and furanone compounds), were detected, which is the highest number of VOCs that have ever been detected in strawberries. The sensory evaluation results indicated that there was a notable influence of imidacloprid (IMI) on the aroma of the strawberries. Principal component analysis and partial least squares discriminant analysis results suggested that the composition of volatile compounds significantly differed in the present study between the IMI-treated and non-IMI-treated groups. Furthermore, the flavor-related indicators of 25 key contributors to the differences between the two treatment groups suggested that VOC profiles can be considered an indicator for distinguishing between strawberries from different agricultural practices. Flavoromics can provide new insights into the quality of strawberries from different agricultural practices.

## 1. Introduction

Strawberries (Fragaria × ananassa) are a valuable fruit, cultivated worldwide, and provide consumers with a rich sensory experience and antioxidants [1]. In 2021, the global Gross Production Value of strawberries was over US$ 22.87 billion, 7.68–37.85 times higher than that of raspberries, blueberries, and cranberries [2]. In recent years, organic farming has flourished due to prohibitions of synthetic fertilizers and pesticides, and is friendly to ecosystems, animals, and humans. Approximately 88.15% of 559 valid questionnaires agreed that organic products were better than conventional ones [3]. Akšić et al. [4] found that there was a significant difference (*p* < 0.05) in carbohydrate content between integrated (15.77 mg/g) and organic (10.67 mg/g) strawberries. The anthocyanin content is also significantly lower (*p* < 0.05) in strawberry puree from conventional cultivation systems (174.11 mg/kg) than in strawberry puree from organic cultivation systems (291.24 mg/kg) [5]. Furthermore, higher ascorbate to dehydro-ascorbate ratios, total phenolic contents, and greater antiproliferative activities were detected in organically cultivated strawberries than in conventionally grown strawberries [6]. However, there is still limited knowledge of how organic practices may affect the abundance and composition of aroma components in strawberries. Flavor is one of the primary drivers of consumer preferences for strawberries [7]. Does the organic agricultural (OA) production mode increase the flavor of strawberries in comparison with conventional agriculture (CA)? More evidence is needed to fill this knowledge gap.

The flavoromics strategy has been collectively considered, and provides new perspectives on the correlation between the aroma of foods and their chemical profiles [8], which can help us understand VOC profiles comprehensively. At the time of writing, a total of 979 VOCs have been detected in strawberries by gas chromatography-flame ionization detection (GC-FID), gas chromatography−mass spectrometry coupling (GC-MS), gas chromatography-ion mobility spectrometry (GC-IMS), and multidimensional gas chromatography (MDGC-MS) [9]. Among these techniques, MDGC-MS detected the highest amount simultaneously (567 VOCs) [10]. GC×GC-TOF-MS has been successfully used to identify aroma compounds in agro-products due to its unparalleled separation ability. Zhang et al. [11] identified 211 volatile components in green tea using GC×GC-TOF-MS, which is approximately 6.4 times more than that identified by GC-IMS. A total of 126 volatile components were detected in nuts by GC×GC-TOF-MS, approximately three times higher than that identified by GC-MS [12]. However, no information is available on the influence of pesticides on the VOC profile of strawberries. One article has reported 852 VOCs detected by GC×GC-TOF-MS in the ripe strawberry fruit under three preservative treatments, including 298 unknown compounds [13]. According to the previous study, the insecticide imidacloprid is widely used in China to protect strawberries from aphids [14], a major pest of strawberries [15]. Previous studies have reported that imidacloprid could trigger metabolic disturbances (e.g., in amino acid metabolism and saccharide catabolism) in chickpeas [16], cucumbers [17], and tomatoes [18]. However, to our knowledge, the effect of imidacloprid on the VOCs of plants has only been observed in thyme [19]. Considering the importance of VOCs to aroma and flavor perception, there is much to be further investigated regarding the VOC profiles of strawberry fruits. 

Therefore, in the present study, the components and profiles of VOCs in the ripe strawberry fruit under organic (non-pesticide-treated) and conventional (pesticide-treated) cultivation practices were identified and characterized by GC×GC-TOF-MS. Furthermore, the volatile components possibly accounting for the aroma characteristics of the strawberries under different agricultural practices were explored and discussed. The results are expected to provide insights into the flavor qualities of strawberries under different agricultural practices, based on flavoromics.

## 2. Materials and Methods

### 2.1. Sample Treatment

Field experiment: Strawberry (Fragaria × ananassa Duch. cv. ‘Benihoppe’) treatments were conducted in an organic greenhouse from March 2022 to May 2022 in the Hongnile modern agricultural farm, Changping District, Beijing, China (116°09′ E, 40°13′ N). ‘Benihoppe’ is the most important strawberry cultivar grown in China. Day/night temperatures and relative humidities averaged 89.6 °F/64.4 °F (32 °C/18 °C) and 35%/80%, respectively. The organic and conventional agricultural practices in this work mainly differed with regards to pesticide control. The strawberry plants were assigned to two groups with an interval of five meters when fruit setting. The first group (control group) was sprayed with distilled water. The second group (IMI) was treated with the recommended dose (374.8 g/hm^2^) of imidacloprid. The two treatment groups were treated with imidacloprid and distilled water once during the growing season [20]. Both groups consisted of three replicated plots. After an exposure period of 20 days at the commercial ripening stage, six strawberry fruits were randomly selected from each plot in the two groups. These samples were rapidly frozen using liquid nitrogen, immediately transported to the laboratory, and subsequently homogenized. The homogenized fruit samples were then stored at −80 °C until they were ready for analysis.

### 2.2. GC×GC-TOFMS Analysis

GC conditions: carrier gas—high-purity helium (99.999%); flow—1 mL/min; split ratio—3:1; injector and transfer line temperatures—250 °C; modulation column—HV column (1.3 m × 0.25 mm I.D.×0.25 µm; Hexin Analytical Instrument Co., Ltd., Guangzhou, China); modulation period—7 s; the first-dimensional column—DB-5MS column (30 m × 0.32 mm I.D.×0.25 µm; Agilent Technologies, Santa Clara, CA, USA); temperature procedure—40 °C for 2 min, increased to 200 °C at 2 °C/min, finally ramped up to 270 °C at 6 °C/min, and held for 5 min; the second-dimensional column—DB-17 column (2 m × 0.25 mm I.D.×0.15 μm; Agilent Technologies, Santa Clara, CA, USA); oven offset temperature—+0 °C relative to the GC oven; modulator temperature offset—+30 °C relative to the secondary oven.

MS conditions: ion source temperature—250 °C; mass range—10–550 *m*/*z*; electron ionization (EI) mode—70 eV; detector voltage—−1730 V.

### 2.3. Qualitative Analysis and Quantitation Assurance

Calibration curves with seven calibration points were built to cover analyte amounts in the strawberry samples, with a range of 0.01–1 µg/g for 2-heptanone, benzaldehyde, 1-heptanol, 1-decene, ethyl hexanoate, DMMF, eugenol, and 1-teradecanol. The coefficients of determination (R^2^) were within the range of 0.9297–0.9923, illustrating the reliability of the method (Table 1 and Appendix A). Recovery experiments were carried out on strawberry samples at 0.1–0.5 µg/g with standards of 2-heptanone, benzaldehyde, 1-heptanol, 1-decene, ethyl hexanoate, DMMF, eugenol, 2-heptanone, and 1-teradecanol. Three repetitions of recovery and repeatability data were used to evaluate trueness and precision. As shown in Table 1, the recovery values for spiking the strawberry with eight external standards were 64.76–132.48%, with relative standard deviations (RSDs) of 0.48−21.17%. The limits of detection (LODs) and limits of quantification (LOQs) were calculated based on signal-to-noise (S/N) values of 3 and 10, which were in the range of 0.024–0.13 and 0.08–0.55 ng/g for the strawberry samples, respectively.

### 2.4. Sensory Evaluation and Statistical Analysis

Twenty-six assessors (16 females, 10 males) were assigned to sensory evaluation. The assessors were asked to evaluate the mature strawberries from two control groups and one IMI-treated group. The calyxes of fresh strawberries were removed and rinsed in water. In the sensory evaluation, the assessors rinsed their mouths with mineral water between samples, and no additional information was given to the assessors [21]. Samples were coded with three random digits. No information on sample order was provided. Based on a previously described scoring method, ‘strawberry-like aroma intensity’, ‘fresh aroma intensity’, ‘sweetness intensity’, ‘sourness intensity’, and ‘smelly intensity’ were evaluated [22]. 

## 3. Results and Discussion

### 3.1. GC×GC–TOFMS Topographic Plots of Strawberries under Different Treatments

The three-dimensional (3D) spectra obtained from GC×GC–TOFMS analysis are shown in Figure 1. In both treatments, most of the signals appeared in 1D-RT of 2.0–60.0 min and 2D-RT of 0.0–7.0 s. The number and relative content of volatile components in organic strawberry samples (without IMI, Figure 1A) were lower than in IMI-treated strawberry samples (Figure 1B) between 20.0 and 60.0 min of 1D-RT, and between 2.0 and 7.0 s of 2D-RT. Similar phenomena were also reported in a previous study [19], possibly due to photochemical reactions between IMI and the volatile compounds of strawberries, resulting in the release of the radical NO_2_ under irradiation. These results indicate that IMI treatment affected the volatile compounds of strawberries.

### 3.2. Identification of Volatile Components in Strawberries under Different Treatments

A total of 1164 volatile compounds (Appendix A) were tentatively identified in the present study, more than have been previously detected in strawberries (979), as reviewed by Ulrich et al. [9]. The identified volatile components were classified into 23 categories in the present study (Figure 2), including esters (254), alcohols (130), ketones (101), aldehydes (77), acids (62), and others. The major compounds of esters, ketones, aldehydes, terpenes, and furanones were identified in the strawberry samples; the chromatograms and mass spectra are provided in Appendix A. The most abundant five categories were identified as the predominant flavor components of strawberries, as described in previous studies [23,24,25]. Moreover, 354 compounds were identified in strawberries for the first time compared with previous reports [9,13]. The most numerous chemical groups were the esters (61), followed by ketones (33), alkenes (33), and aldehydes (26). The results suggested that the separation ability of GC×GC-TOFMS has significant qualitative advantages. 

Figure 3A shows a slightly higher total number of VOCs identified in IMI-treated strawberries (825) in comparison with the non-IMI group (779), including 440 compounds that were found in both. The number of VOCs observed in each treated group was higher (553) than in strawberries previously observed by Cannon et al. [10]. Furthermore, the combination of the most abundant five categories of aldehydes, esters, alcohols, acids, and ketones, exceeded 50% in both non-IMI and IMI-treated strawberries (Figure 3B). This result was consistent with Zhao et al. [24], where they accounted for 60% of the total. The number of aldehyde compounds under IMI treatment (68) was approximately the same as in non-IMI-treated (61) strawberries. Significantly, esters were the most numerous compounds, with higher numbers under IMI treatment (198) than in the non-IMI group (163), consistent with results reported previously in the literature [10,26]. Interestingly, IMI treatment also increased the number of alcohol and ketone compounds. This result might be due to the increased number of aldehydes, which are precursors of alcohols and ketones in strawberries, via LOX and hydroperoxide lyase (HPL) [27,28]. Similar synthetic pathways were also reported under IMI treatment in tea plants [29]. These results suggest that IMI treatment may influence the LOX-HPL pathway in strawberries. In addition, the application of IMI increased the number of acids from 37 to 51. Sharma et al. [30] and Zhou et al. [31] proposed that the accumulation of organic acids could defend against stress caused by IMI and other neonicotinoid insecticides. These results indicated that IMI triggered the stress response and interfered with the VOC composition of the strawberries.

### 3.3. Profile of the Volatile Components in Strawberries under Different Treatments

Figure 4 presents the peak area profiles of the volatile compound categories for the two treatment samples. The total volatile peak area was increased 1.5-fold in the IMI-treated samples relative to the non-IMI-treated samples. The main contributions to the peak areas of the two groups were from aldehydes, esters, alcohols, acids, and ketones, and accounted for more than 90% of the areas of both treatment groups. The five types of compounds mentioned above have been reported to be critical factors contributing to the fruit aroma of the ‘Benihoppe’ strawberry [32,33]. Aldehydes were the predominant compounds in the non-IMI and IMI-treated strawberry samples, which was in keeping with the previous report [34]. Yang et al. [35] found that in postharvest strawberries in the control and 1-methylcyclopropene-treated groups, aldehydes accounted for 84.53% and 86.63%, respectively. In comparison with the non-IMI group, significantly higher (*p* < 0.05) peak areas for aldehydes, esters, acids, ketones, and furanone compounds were found in the IMI-treated group. It has been reported that aldehydes, esters, alcohols, and ketones are mainly metabolized by the fatty acid and amino acid pathways in berries [36]. Lipoxygenase (LOX) catalyzes the oxygenation of polyunsaturated fatty acids, and was reported to be a key enzyme in the production of VOCs in fruits. Similarly, Hamilton-Kemp et al. [30] have proved that IMI affects the transcript level of the *CsLOX3* gene, which was classified as the gene encoding LOX. Therefore, IMI treatment could further affect the synthesis of aldehydes, esters, alcohols, and ketones.

***Aldehydes*** Aldehydes are mainly associated with fresh, green notes in strawberries. Hexanal, octanal, nonanal, and (E)-2-hexenal were the primary aldehydes in non-IMI and IMI-treated strawberries, accounting for 77.51% and 68.22% of the total aldehydes, respectively (Figure 5A). Several pieces of research have confirmed that of the predominant compounds in strawberries, hexanal was responsible for the fruity notes [7,36]. A higher peak area for hexanal was found in non-IMI-treated strawberries (26.22%) than with IMI treatment (18.64%). In addition, benzaldehyde appeared only in IMI-treated strawberries (6.15%) and was identified as an off-odor aroma compound in strawberries [7]. Misran et al. [37] found that during storage, hexanal was degraded while benzaldehyde was formed, which negatively influenced the aroma of the stored strawberries.

***Esters*** Esters represent the primary source of fruity and floral odors in strawberries. The distribution differences for specific ester compounds are clearly shown in Figure 5B. Twenty ester compounds accounted for 90% of the total esters under non-IMI treatment. Methyl hexanoate was the most predominant ester in non-IMI-treated strawberries (21.55%) with a sweet flavor, in agreement with the previous observation of pineapple [35]. This was followed by ethyl hexanoate (16.69%) and methyl butyrate (11.45%). These proportions were approximately 2–4 times higher than in the IMI-treated strawberries, which suggested that IMI treatment can influence the composition of esters. Unlike the non-IMI treatment group, the top three highest percentages of total peak areas for ester compounds in IMI-treated strawberries were for ethyl butyrate (32.31%), hexyl acetate (12.80%) and ethyl hexanoate (9.62%). Of these compounds, ethyl butyrate was detected only in the IMI-treated strawberries, and has been identified as one of the key constituents of the fresh strawberry aroma [38]. Ethyl butyrate has a natural pineapple-like aroma, and is frequently found in fruits such as mango and grape. However, several previous studies reported that the content of ethyl butyrate was reduced to below the detection limit of GC-MS and GC×GC-TOFMS in frozen (below −20 °C) strawberry samples [39,40,41]. Thus, these results indicated that IMI treatment might significantly increase the content of ethyl butyrate.

***Alcohols*** There was little influence of IMI-treatment on the profile of alcohol compounds (Figure 5C). 2-Hexen-1-ol and 1-octanol were the most abundant alcohols in non-IMI and IMI-treated strawberries, accounting for 73.75% and 79.38%, respectively. Octacosanol, which was reported as the prominent aliphatic primary alcohol in all policosanol mixtures, was first detected in strawberries and identified only in non-IMI-treated samples (4.70%) in the present study. On the contrary, (R)-Oxiranemethanol was detected only in IMI-treated strawberries (2.66%). This compound has been recognized as a probable human carcinogen in vaping products, resulting from the thermal degradation of glycerol [41], and was included in the *Harmful and Potentially Harmful Constituents List* in 2019 [42]. (R)-Oxiranemethanol is primarily found in refined fats, oils, and foods containing fats and oils [43,44]. In comparison, its methyl derivative was found in green tea, and strawberries with a green odor. However, (R)-Oxiranemethanol can be rapidly hydrolyzed to glycerol, and then reacts readily with glutathione to form S-(2,3-dihydroxy propyl) glutathione. Therefore, we inferred that the appearance of (R)-Oxiranemethanol in IMI-treated strawberries was due to IMI restraining the rapid metabolic pathway of (R)-Oxiranemethanol.

***Ketones*** Ketones are mainly associated with creamy notes in strawberries. Thirteen and eight compounds accounted for 90% of the total ketones of the IMI and non-IMI-treated groups, respectively. Consistent with previous reports, 2-heptanone appeared to be the most abundant ketone in both groups, more than 1.7-fold higher in the non-IMI-treated (54.22%) strawberries (Figure 5D), followed by acetone. 2-Heptanone was reported to be one of the major constituents giving the strawberry aroma a fruity and fresh odor [45]. Methyl isobutyl ketone was found only in the non-IMI treatment group, and accounted for 5% of the total ketone compound peak area. Previous research demonstrated that methyl isobutyl ketone significantly (VIP value > 1) correlated with the consumer liking tendency, with a fruity and ethereal odor [7]. In contrast, as shown in Figure 5D, 3-octanone (11.00%) was identified only in IMI-treated strawberries with fruity and cheesy aromas. In agreement with these results, Jelen et al. [46] have found that 3-octanone was detected only in strawberries under biotic stress. Although no further information has been reported on the mechanism of this phenomenon, we suspected that 3-octanone was involved in the biotic and abiotic stress responses of strawberries.

***Others*** Since terpenes and furan have characteristic sensory properties, these compounds are particularly interesting in strawberries. In the current study, a total of 39 terpene compounds were detected in all of the analyzed strawberry samples (Figure 5E), which was higher than that found by the HS-SPME-GC-FID (4), HS-SPME-GC-MS (10), and HS-GC-IMS (5) methodologies [11,34,41]. In non-IMI treated strawberries, the predominant terpene, linalool, accounted for 62.67% of terpenes, followed by trans-nerolidol (25.66%), and D-limonene (8.93%). In contrast, trans-nerolidol (35.73%) ranked first in the IMI-treated group, followed by nerolidol (31.84%), and linalool (29.10%). Of these terpenes, nerolidol was found only in IMI-treated strawberries with a floral odor. Linalool was a typical aroma in those strawberries with floral and lavender flavors. IMI treatment significantly reduced the linalool content (*p* < 0.05), which indicated that the floral odor of strawberries might be affected by IMI. In addition, 20 furan compounds were detected in all of the analyzed strawberry samples, including eight furanone compounds (Appendix A). Trans-2-(2-pentenyl) furan (49.31%) and 2-pentyl-furan (43.41%), which were first found in strawberries, were the two most predominant furan compounds and were detected only in IMI-treated samples (Figure 5F). Trans-2-(2-pentenyl) furan gives mahaleb cherries a particular aroma with a fruity flavor [47]; 2-pentyl furan was more prevalent in the soybean, roasted nuts, and dried fruits [18,33]. In soybean oil, these were made from polyunsaturated fatty acids through triplet oxygen autoxidation [48], suggesting that the production of trans-2-(2-pentenyl) furan and 2-pentyl furan might be due to lipid oxidation caused by IMI [32]. Furanone was reported as the characteristic strawberry volatile compound offering a strawberry-like odor [49]. According to Figure 5F, IMI significantly (*p* < 0.05) reduced the content of 4-methoxy-2,5-dimethyl-3(2H)-Furanone (DMMF), which is recognized as one of the predominant contributors to the characteristic strawberry flavor. It has been reported that D-fructose and D-fructose-1,6-bisphosphate are the predominant precursors to furanone in the glucometabolic pathway, the amount of which was significantly affected by neonicotinoid insecticides [18,32]. Therefore, the profile results of volatile constituents in the present study indicate that different agricultural practices may contribute to differences in the aroma of strawberries.

### 3.4. Determination of Key Volatile Compounds Accounting for the Aroma Characteristics of Strawberries in the Non-IMI Treatment and IMI Treatment Groups

In order to identify the potential volatile markers of strawberry samples from different treatments, PCA and PLS-DA were performed using the peak areas of the volatile compounds. As shown in Appendix A, strawberries with 12 replicates were collected into each treatment group, and separated in the PCA (Appendix A) and PLS-DA plots (Appendix A), indicating that the composition of volatile compounds varied significantly between the two treatment groups. 

Table 2 lists the 47 VOCs responsible for discrimination between the two groups (VIP value > 1), including 20 esters, 8 aldehydes, 6 ketones, 4 terpenes, 4 alcohols, and 2 furanones, among others. Twenty-five of the forty-seven VOCs had relatively lower odor threshold values, except for esters. Of these VOCs, D-limonene (0.034 mg/kg), (E)-2-hexenal (0.0885 mg/kg), and 2,5-dimethyl-4-methoxy-3(2H)-furanone (DMMF, 0.00003 mg/kg), which have extremely low detection thresholds, were significantly (*p* < 0.05) reduced in strawberries by IMI treatment, whereas methyl salicylate (0.04 mg/kg), 2-nonanone (0.04 mg/kg), and γ-dodecalactone (0.00043 mg/kg) were significantly (*p* < 0.05) increased, and could be used as key biomarkers for distinguishing between the flavors of the two strawberry groups. Aroma description remains fundamental to estimating the contributions of these aroma-active volatile compounds to the overall aroma impression. All seven candidate markers were reported previously for aroma descriptions of fruits, including nona-3,5-dien-2-one (green, grassy), D-limonene (citrus, mint), (E)-2-hexenal (fruity, fresh, green), DMMF (strawberry-like, floral, fruity), methyl salicylate (holly leaf, fresh), 2-nonanone (floral, oils, and herbs), and γ-dodecalactone (peach, milk peach). Among these compounds, the relative contents of D-limonene, (E)-2-hexenal, and DMMF were detected as being higher in non-IMI-treated strawberries with leading ‘strawberry-like’ and ‘fruity’ aromas. At the same time, the other four compounds were higher in IMI-treated strawberries with central ‘green’ and ‘fresh’ aromas. The results of the sensory evaluations of strawberries from the two control groups and the one IMI-treated group are shown in Figure 6. Different tendencies were observed in the results of the control and IMI-treated groups. The ‘strawberry-like aroma’ intensity was higher for strawberries from the control group, while the ‘fresh aroma’ intensity was lower. In terms of overall preference, all 26 assessors preferred non-IMI-treated strawberries. Combining the sensory evaluation results (Figure 6) and the available flavor threshold and description results, as well as the relative contents (Table 2), it is suggested that IMI could have a negative influence on ‘strawberry-like’ aromas and a positive influence on ‘fresh’ aromas, a phenomenon which reduces the popularity of strawberries among consumers. These results could provide basic data for management regulations regarding the use of imidacloprid on strawberries in the future. 

## 4. Conclusions

The present study investigated the distribution of, and differences in, the VOCs in strawberries from different agricultural practices, i.e., organic (without imidacloprid) and conventional (with imidacloprid). A total of 1164 VOCs, representing 23 chemical classes, were detected (354 of these for the first time) by GC×GC-TOFMS, which is the highest number of VOCs that have ever been detected in strawberries. Volatile compositions varied significantly (*p* < 0.05) between the two treatment groups in this study. In comparison with the non-IMI-treated group, the strawberries with the 374.8 g/hm^2^ IMI treatment had higher responses for five characteristic aroma compounds, including aldehydes (31.59%), acids (17.25%), esters (25.01%), ketones (5.32%), and furanones (0.11%), which have important flavor attributes for strawberries and strongly influence consumer acceptability. The results of the sensory evaluations, flavor thresholds, and odor descriptions, along with the relative content of 25 key contributors (VIP value > 1), have suggested that IMI may have a negative influence on ‘strawberry-like’ aromas and a positive influence on ‘fresh’ aromas. It is essential to point out that the vital volatile compounds evaluated in this study represent only a tiny proportion of the total key volatile compounds present in strawberry samples. The results on the volatile constituents in the present study indicated that different agricultural practices might contribute to flavor differences in strawberries. The VOC profile can be considered an indicator for distinguishing strawberries. However, the mechanisms through which pesticides (such as imidacloprid) affect the flavors of strawberries remain unclear. Thus, future work is needed to explore the mechanisms of the interactions between pesticides and strawberry aroma components using proteomic, metabolomic, and transcriptomic approaches. 

## Figures and Tables

**Figure 1 foods-12-02914-f001:**
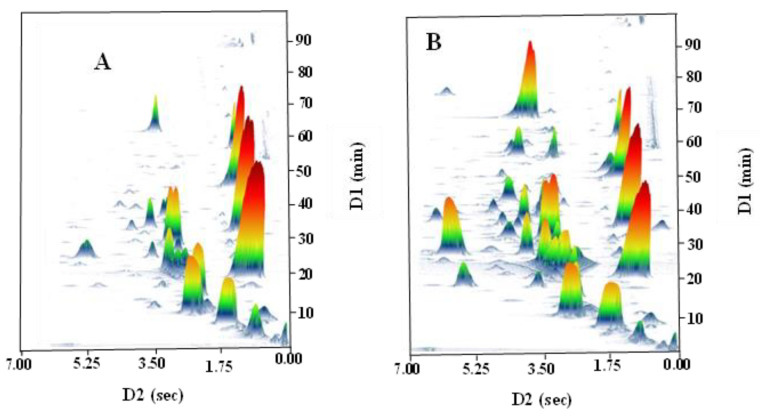
Fingerprints of volatile compounds in non-IMI-treated and IMI-treated strawberries obtained with GC×GC-TOF-MS. (**A**) non-IMI-treated; (**B**) IMI-treated.

**Figure 2 foods-12-02914-f002:**
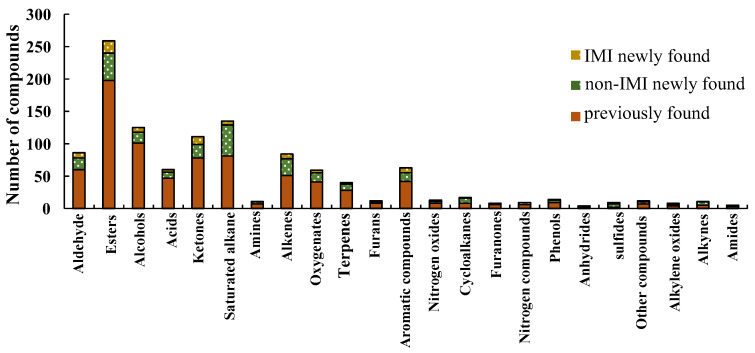
The numbers of compounds in IMI-treated and non-IMI-treated strawberries, obtained by GC×GC-TOFMS.

**Figure 3 foods-12-02914-f003:**
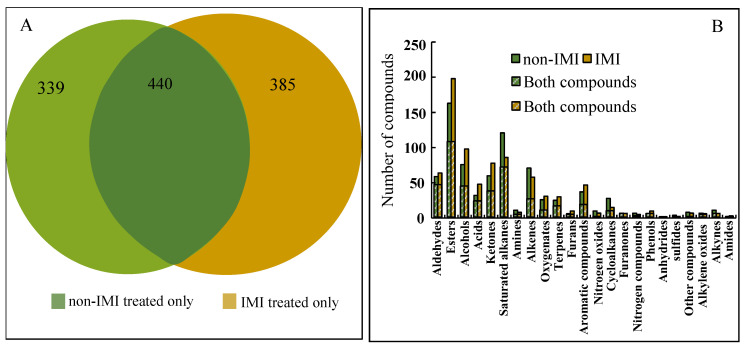
Comparison of the composition of compounds in 23 catalogs with those in IMI-treated and non-IMI-treated strawberries, obtained by GC×GC-TOF-MS. (**A**) Compounds in different treatments; (**B**) compounds in 23 catalogs.

**Figure 4 foods-12-02914-f004:**
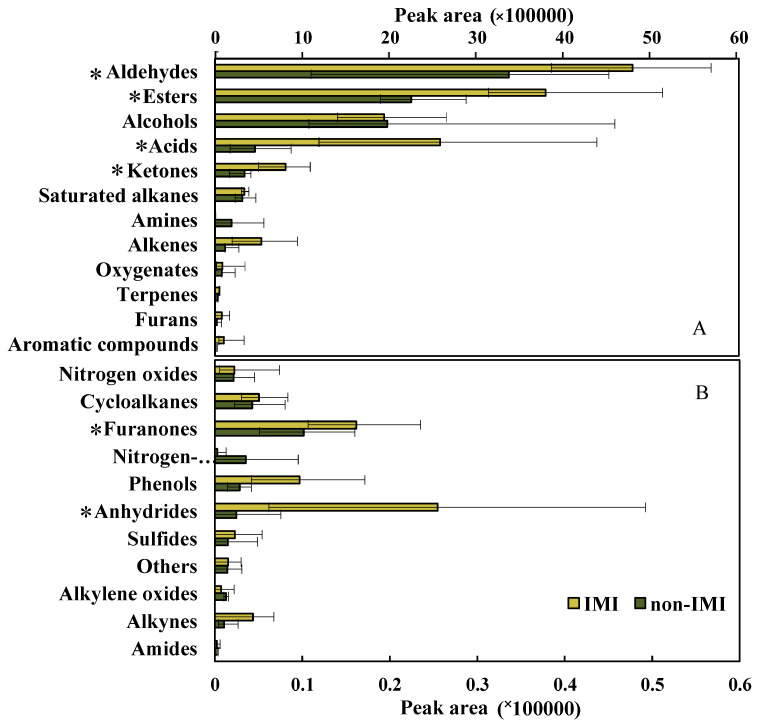
Comparison of the peak areas of compounds in 23 catalogs, for IMI-treated and non-IMI-treated strawberries, obtained by GC×GC-TOFMS. (**A**) peak areas > 0.5 × 10^5^; (**B**) peak areas < 0.5 × 10^5^. * represents a significant difference.

**Figure 5 foods-12-02914-f005:**
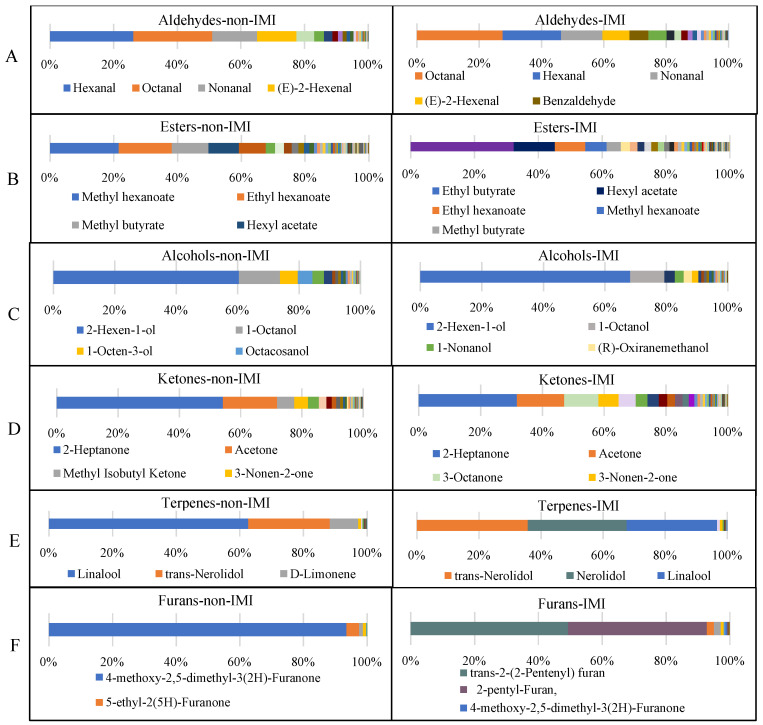
Relative contents of the volatile organic compounds in strawberries from non-IMI treatment and IMI treatment groups. (**A**) Aldehydes; (**B**) esters; (**C**) alcohols; (**D**) ketones; (**E**) terpenes; (**F**) furans.

**Figure 6 foods-12-02914-f006:**
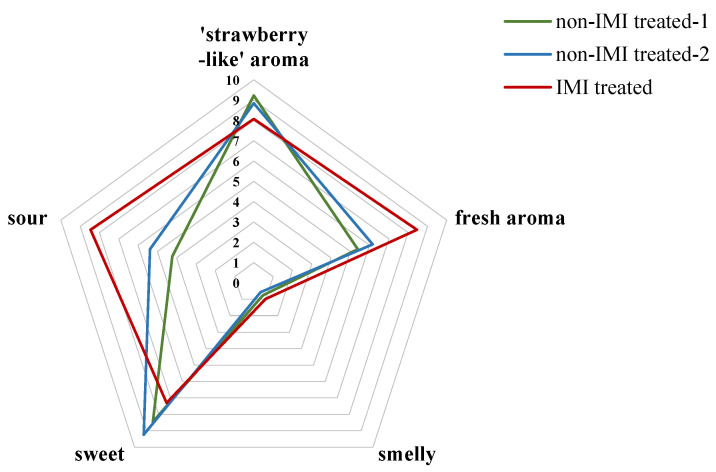
Results of the sensory evaluations of mature strawberries.

**Table 1 foods-12-02914-t001:** The calibration curve and method sensitivity of external standard with GC×GC-TOFMS by headspace SPME.

Compounds	Standard Curve	Correlation Coefficient (R^2^)	LOD (ng/g)	LOQ (ng/g)	Average Absolute Recovery (%)			
					0.5 (µg/g)	0.2 (µg/g)	0.1 (µg/g)	RSD (%)
2-Heptanone	y = 261.48x + 19,966	0.9741	0.024	0.08	91.89	97.12	132.48	4.43–9.09
Benzaldehyde	y = 492.68x + 35,838	0.9777	0.043	0.14	64.76	84.08	101.73	4.74–16.87
1-Heptanol	y = 325.99x + 125.2	0.9916	0.16	0.55	70.54	85.03	103.45	4.97–8.99
1-Decene	y = 469.76x + 56,524	0.9772	0.044	0.15	72.71	108.57	130.47	5.00–14.38
Ethyl Hexanoate	y = 906.61x + 111,217	0.9297	0.069	0.23	79.49	126.98	85.04	0.48–2.88
DMMF	y = 521.59x + 4945.2	0.9923	0.036	0.13	82.73	110.27	89.82	3.52–12.25
Eugenol	y = 610.89x − 6562.2	0.9887	0.13	0.42	69.09	74.17	83.75	11.57–21.17
1-Teradecanol	y = 340.58x − 11,455	0.9685	0.053	0.18	69.64	79.36	96.03	5.83–10.49

**Table 2 foods-12-02914-t002:** GC×GC-TOFMS peak areas (% on the total area) of difference contributors (VIP value > 1) in non-IMI and IMI treated strawberry samples.

Category	Compounds	CAS Number	VIP Value	Relative Content		Aroma Description	Odor Threshold (mg/kg)	Medium
				Non-IMI	IMI			
Esters	Propanoic acid, 2-methyl-, 3-hydroxy-2,2,4-trimethylpentyl ester	77-68-9	1.08	0.015 ± 0.001	0.016 ± 0.001		0.58 d	air
	Methyl 3-hydroxyoctanoate	7367-87-5	1.15	0.011 ± 0.003	0.055 ± 0.006			
	3-Methylheptyl acetate	72218-58-7	2.02	0.011 ± 0.002	0.012 ± 0.006			
	Methyl 3-hydroxytetradecanoate	55682-83-2	1.00	0.011 ± 0.002	0.030 ± 0.007			
	Pentyl hexanoate	540-07-8	2.16	0.009 ± 0.001	0.039 ± 0.004	floral, fresh, rose		
	Octyl 2-methylbutyrate	29811-50-5	1.00	0.017 ± 0.003	0.011 ± 0.002			
	Butanoic acid, hexyl ester	2639-63-6	1.58	0.547 ± 0.046	0.615 ± 0.022	pineapple	0.203 d	water
	S-Methyl 3-methylbutanethioate	23747-45-7	1.02	0.053 ± 0.006	0.037 ± 0.011			
	Methyl 3-hydroxyhexanoate	21188-58-9	1.02	0.029 ± 0.004	0.067 ± 0.006		0.05	oil
	4-Octenoic acid, methyl ester	1732-00-9	1.28	0.053 ± 0.014	/			
	Acetic acid, phenylmethyl ester	140-11-4	2.15	0.140 ± 0.014	0.442 ± 0.046	floral		
	Methyl salicylate	119-36-8	2.24	0.087 ± 0.017	0.388 ± 0.026	holly leaf, fresh	0.04	water
	2-Buten-1-ol, 3-methyl-, acetate	1191-16-8	1.41	0.013 ± 0.004	0.032 ± 0.015			
	Acetic acid, octyl ester	112-14-1	1.43	0.124 ± 0.038	0.224 ± 0.047		0.047 d	water
	Acetic acid, heptyl ester	112-06-1	1.47	0.017 ± 0.004	0.050 ± 0.007		0.42 d	water
	Methyl hexanoate	106-70-7	0.88	4.694 ± 1.934	2.389 ± 0.492	fruity, pineapple	0.07	water
	Hexanoic acid, 2-methylpropyl ester	105-79-3	1.07	0.008 ± 0.002	0.002 ± 0.001			
	Ethylene glycol di-n-butyrate	105-72-6	1.11	0.062 ± 0.021	0.030 ± 0.005			
	Acetic acid, 2-phenylethyl ester	103-45-7	1.41	0.013 ± 0.002	0.034 ± 0.004		0.24959 d	water
	Butanoic acid, decyl ester	1298317	1.06	0.002 ± 0.001	0.042 ± 0.004			
Ketones	Methyl nicotinate	93-60-7	2.05	0.122 ± 0.023	0.331 ± 0.100			
	2-Nonanone	821-55-6	1.27	0.078 ± 0.016	0.231 ± 0.017	floral, oils and herbs	0.04 d	water
	Nona-3,5-dien-2-one	80387-31-1	1.36	0.070 ± 0.015	0.225 ± 0.058	green grassy		
	1-Octen-3-one	4312-99-6	1.20	0.027 ± 0.004	0.073 ± 0.010	soap, gasoline	0.000003 d	water
	2-Undecanone	112-12-9	1.30	0.005 ± 0.002	0.030 ± 0.002	green, citrus, fresh	0.0055 d	water
	γ-Dodecalactone	2305-05-7	1.20	0.024 ± 0.005	0.310 ± 0.043	peach, milk peach	0.00043 d	water
Terpenes	D-Limonene	5989-27-5	2.51	0.399 ± 0.169	/	citrus, mint	0.034 d	water
	β-Cyclocitral	432-25-7	1.17	0.049 ± 0.011	0.065 ± 0.004	tobacco, floral	0.003	water
	β-Myrcene	123-35-3	2.53	0.005 ± 0.002	0.017 ± 0.006	sweet orange, balsam	0.0012 d	water
	nerol	106-25-2	1.22	0.007 ± 0.001	0.012 ± 0.001	lemon	0.68 d	water
Acids	Acetohydroxamic acid	546-88-3	1.29	0.015 ± 0.007	/			
Aldehydes	(Z)-4-Heptenal	6728-31-0	1.05	0.004 ± 0.001	0.003 ± 0.002			
	(E)-2-Hexenal	6728-26-3	1.08	2.963 ± 2.687	1.732 ± 1.576	fruity, fresh, green	0.0885 d	water
	(E,Z)-2,6-Nonadienal	557-48-2	1.11	0.471 ± 0.099	0.389 ± 0.026	cucumber	0.0008 d	water
	Benzaldehyde, 4-ethyl-	4748-78-1	1.00	0.006 ± 0.003	0.007 ± 0.004	bitter almond	0.013 d	air
	(Z)-2-Decenal	2497-25-8	1.35	0.081 ± 0.007	0.102 ± 0.023	tallow, waxy, fatty		
	Benzeneacetaldehyde	122-78-1	1.08	0.122 ± 0.020	0.183 ± 0.022	hyacinth, almond and cherry	0.004 d	water
	Decanal	112-31-2	1.38	0.530 ± 0.113	0.481 ± 0.044	thrill	0.003	water
	4-Oxononanal	74327-29-0	1.22	0.006 ± 0.001	/			
Cycloalkanes	1,7-Dimethyl-4-(1-methylethyl)cyclodecane	645-10-3	1.98	0.009 ± 0.004	/			
Furanone	5-(acetyloxy)dihydro-5-methyl-2(3H)-furanone	57681-51-3	1.05	0.002 ± 0.001	/			
	2,5-Dimethyl-4-methoxy-3(2H)-furanone (DMMF)	4077-47-8	1.51	2.157 ± 0.291	0.022 ± 0.007	strawberry-like, floral, fruity,	0.00003 d	water
Alcohols	2-Hexen-1-ol	56922-75-9	1.04	2.468 ± 0.720	4.343 ± 1.994			
	1-Nonanol	143-08-8	1.46	0.344 ± 0.071	0.365 ± 0.019	sweet orange, sweet and green rose	0.001	
	1-Decanol	112-30-1	1.54	0.009 ± 0.001	0.082 ± 0.005	fatty	0.0066 d	water
	1-Heptanol	111-70-6	1.02	0.040 ± 0.007	0.077 ± 0.007	fat, pungent, citrus	0.0054 d	water
	3-Nonen-1-ol, (Z)-	10340-23-5	1.63	0.007 ± 0.001	0.012 ± 0.003			

## Data Availability

The data used to support the findings of this study can be made available by the corresponding author upon request.

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
