# Peer review of "The Effect of Imidacloprid on the Volatile Organic Compound Profile of Strawberries: New Insights from Flavoromics"

_foods, 2023, doi:10.3390/foods12152914_

Round 1

Reviewer 1 Report

Manuscript ID: foods-2500521-peer-review-v1

 The topic “New insights from flavoromics on strawberries: different vola-tile organic compounds profile under organic and conventional agricultural practices” is an interesting and novelty in the field. However, some small points of the manuscript need to revise.

Minor revision is required.

Comments are below…

1.     Typographical errors exist throughout the manuscript. Rectify carefully.

2.     The plagiarisms in the whole manuscript need to revise. The level of similarity seems too high (Almost 24%). The PDF file for similarity checking is attachment file, please check it out. Author need to concern to revise this point.

3.     Figure 5 need to increase the resolution.

4.     Based on the current study, what is the benefit for take or utilize this data from the experiment in the real situation for fruit criteria? If the focus in term of the food chain.? Please justify.

5.     In the conclusion seems to be in general and is not given separately, it is highly recommended to include limitation of the study and potential future research goals.

None.

Reviewer 2 Report

see report

see report

Reviewer 3 Report

2.1.   Regarding the biological material used. You mentioned that the strawberry fruits were grown in a greenhouse 5 meters apart. In this situation how do you distinguish between organic and conventional? Was this greenhouse used for organic cultivation? Is this greenhouse certified for organic cultivation? And if it is then how did you treat with imidacloprid? Is the 5-meter spacing between variants considered safe so that the applied treatment does not interfere with the organic variant?  I think that the two experimental variants should have been grown in different greenhouses: one certified for organic crops and one for conventional crops. For these reasons, strawberries of the first variety cannot be included in the category of organic crops.  I believe that the term ”organic” should be replaced by another terminology.

Please correct the sentence: The second group (IMI) was treated with the recommended dose (374.8 g/hm²) with: The second group (IMI) was treated with the recommended dose (374.8 g/hm²) of imidacloprid.

Please specify how many treatments you carried out during the growing season.

Results and discussions: Please rearrange Figure 4. Also please mention each time the catalog(s) with which you made the comparisons. Stop using the notion of: different catalogs.

I think that table S1 and figure S4 should be included in the paper.

Can you restructure the chapter: Conclusions by adding some data (values, concentrations) obtained?

Reviewer 4 Report

The authors attributed a misleading title to the article. What they tested was the effect of imidacloprid, the only factor that was different in the two sets of plants. Now, organic production differs from conventional production by a number of factors, such as fertilization, all phytosanitary treatments and others. So they are very different comparisons, the one in the title of the article and the one that was actually made. Therefore, the title of the article should be changed to match the content of the article.

The introduction presents strawberry production data for 2020. Since we are in the second half of 2023, there are more recent data that could be presented.

The field trial was based on two sets of plants separated by about 5 meters. There were thus no repetitions in the field. We know that the microclimate inside a greenhouse varies greatly. The distance from the side panels can cause the thermal regime and humidity to be different at different points in the greenhouse. Thus, the observed differences in fruit characteristics could be due to these differences in microclimate, so the conclusions of the article should reflect this circumstance.

It is unclear whether imidacloprid was applied once or multiple times. It looks like it was only once, but the doubt remains.

The study is only of interest to countries where imidacloprid is authorized in strawberry. In many countries (eg all EU countries) imoidacloprid has been banned from all crops due to its effect on bees.

Figure 1 needs to be improved. It is necessary to remove the paragraph symbols. The vertical scale is not linear, it has values with precision to the hundredth (absolutely unnecessary in such high values).

Figure 3A also has the paragraph symbols.

The organoleptic analyzes of the fruits, to verify whether the different composition of volatiles is reflected in a different appreciation by the tasters, is not presented in the article, but is referred to in detail in the conclusions. There seems to have been no statistical treatment of the results of the organoleptic tests and since these were presented in a supplement, it does not seem reasonable to discuss these results in the article's conclusions.

Even though I am not a native English speaker, it seems to me that some terms are used incorrectly. For example, "discrepancies", right in the abstract. I think authors should also review the text for language correctness.

Round 2

Reviewer 4 Report

Comment 1.

The authors insist on keeping the title of the article that does not correspond to its content. The arguments they use to keep the title are not convincing.

Instead of “New insights from flavoromics on strawberries: different volatile organic compounds profile under organic and conventional agricultural practices.” I propose the following: “Effect of imidacloprid on volatile organic compounds profile of strawberry. New insights from flavoromics.”

Comment 2.

Thanks. The authors corrected the text as requested.

Comment 3.

The authors' response confirms that the test carried out had no experimental replications. A single set of plants treated with imidacloprid was compared with a set of plants not treated with this insecticide. The authors cannot guarantee that the microclimate conditions to which the two sets of plants were subjected were the same. The authors claim that the product applied to a set of plants did not affect the control set. This assertion is credible, but that was not what we had identified as the shortcomings of the essay.

Comment 4.

The authors' response is satisfactory and the added text is enlightening enough for the readers.

Comment 5.

The authors' response shows that they know that the study they carried out is important for a number of countries, including their country, China, but, in fact, the study is not important for a wide range of countries. The general trend is that this insecticide will be banned in some of the countries where it still applies. Study is of economic importance for one part of the world, that is, not for the whole world.

Comments 6 and 7.

The authors corrected the graphs as proposed.

Comment 8.

The authors have integrated a new graphic into the text, which was previously in the supplementary material. This chart has a small error. The word "fresh" has an extra "a", making it "freash". The caption text is not properly formatted and there is a line of text between the figure and the figure caption.

Comment 9.

The authors made the requested change.

No additional  comments
